# Source identification of infectious diseases in networks via label ranking

**Jianye Zhou[1], Yuewen Jiang[2]\*, Biqing Huang[1]\***

**1** Department of Automation, Tsinghua University, Beijing, PR China, **2** Clinical College of Chinese Medicine, Hubei University of Chinese Medicine, Wuhan, Hubei, PR China

\* workersky1980@hbtcm.edu.cn (YJ); hbq@tsinghua.edu.cn (BH)

**Data Availability Statement:** All relevant data are within the manuscript and its Supporting information files.

**Funding:** National Science and Technology Major Project of China under grant 2018ZX10201002-004-002. The funders had no role in study design,

## Abstract

### Background

Outbreaks of infectious diseases would cause great losses to the human society. Source identification in networks has drawn considerable interest in order to understand and control the infectious disease propagation processes. Unsatisfactory accuracy and high time complexity are major obstacles to practical applications under various real-world situations for existing source identification algorithms.

### Methods

This study attempts to measure the possibility for nodes to become the infection source through label ranking. A unified Label Ranking framework for source identification with complete observation and snapshot is proposed. Firstly, a basic label ranking algorithm with complete observation of the network considering both infected and uninfected nodes is designed. Our inferred infection source node with the highest label ranking tends to have more infected nodes surrounding it, which makes it likely to be in the center of infection subgraph and far from the uninfected frontier. A two-stage algorithm for source identification via semi-supervised learning and label ranking is further proposed to address the source identification issue with snapshot.

### Results

Extensive experiments are conducted on both synthetic and real-world network datasets. It turns out that the proposed label ranking algorithms are capable of identifying the propagation source under different situations fairly accurately with acceptable computational complexity without knowing the underlying model of infection propagation.

### Conclusions

The effectiveness and efficiency of the label ranking algorithms proposed in this study make them be of practical value for infection source identification.

data collection and analysis, decision to publish, or preparation of the manuscript.

**Competing interests:** The authors have declared that no competing interests exist.

# Introduction

Nowadays, propagation on complex networks has been attracting attention of a large number of researchers. Infectious diseases spread through physical contact network and make large populations suffer from illness [1, 2]. Similar processes take place in computer networks [3] and social networks [4, 5]. The reverse problem of propagation named *source identification* is drawing more interest recently in order to control the propagation effectively.

The original source identification issue can be quite challenging. Direct maximum likelihood (ML) or maximum a posterior (MAP) estimation of the source is difficult to solve as it has been proved to be #P-complete [6, 7].

The propagation process of pathogens, as well as rumors or computer viruses, would be associated with three essential elements, which are network topology $G(V, E)$, propagation model and the propagation source $s$. $G(V, E)$ refers to a static contact network with node set $V$ and edge set $E$. Each node in $V$ represents an individual while each edge in $E$ represents some sort of physical contact or connection which allows pathogens to propagate in the network. Various propagation models have been proposed in order to describe the transmission of contagion through individuals, including Susceptible-Infected model (SI) [8], Susceptible-Infected-Recovered model (SIR) and Susceptible-Infected-Susceptible model (SIS) [9], etc. Different propagation models depict individuals' change of state during the transmission process in the form of automata.

As the reverse problem of propagation, source identification in networks has been actively studied since [3]. Researchers have proposed a variety of algorithms aiming to locate the propagation source under various situations where different categories of observations $O$ are given [10], including complete observation [3, 7, 11–13], snapshot [14–18] and sensor observation [5, 19–22]. The complete observation or snapshot vector $O$ would provide the state of all nodes or some of them on network $G(V, E)$ while sensor observation would concentrate on several sensors which could provide more information containing state, infection time and infection direction.

While some researches attempt to locate the source by estimating $P(s|O)$ or $P(O|s)$ through belief propagation [17, 18] or Monte-Carlo simulation [23], most existing algorithms utilize certain statistical properties as alternatives of direct ML or MAP to infer the propagation source heuristically [3, 7, 11, 14, 16, 24]. These researchers assume that the source would be the center of the infection subgraph which was spanned by the pathogen diffusion from the source. Although these network centrality measures could perform well in certain scenarios [24, 25], we believe that centrality of the propagation source based on the infection subgraph is far from enough. The importance of uninfected nodes in source identification tasks has been exploited and verified [12, 26, 27]. Instead of concentrating on only the infected ones, we make use of the massive highly accessible and easily overlooked uninfected nodes in networks, which can be rather valuable to source identification, to generate the complete observation and snapshot, as is explained later.

Time complexity of most former solutions for source identification in networks ranges from $O(n \log n)$ to $O(n^3)$ [10], where $n$ denotes the number of vertices. Large scale of real-world networks would limit the practicability of these inference algorithms greatly. Also, Complete observation of the whole contact network is obviously difficult to be obtained in real life. Algorithms which can deal with the situation under incomplete observation within relatively short run time would possess more practical value. However, existing methods have some limitations more or less due to more required parameters [17, 18, 23, 28], the high computational complexity [16–18] or particularity of propagation model [5, 19]. Therefore, a novel two-stage framework via label ranking has been proposed to deal with the source identification problem

based on snapshot with acceptable computational complexity which has an approximate linear relationship with $m$, the number of edges.

In this paper, we exploit the potential property of the propagation source as an alternative to identify it effectively. A unified label ranking framework is designed to identify the infection source with complete observation and snapshot via label propagation and semi-supervised learning. Based on the idea of source prominence from [27], we get back to the spreading process in the contact network. The source would infect the nodes surrounding it first. Pathogens then spread to more nodes beyond to generate an infection subgraph. This makes the source be likely to locate in the center of infection subgraph and stay far from the uninfected frontier. Based on this, we intend to measure the possibility for nodes to become the source via *label ranking*. The label ranking score of each node is designed to measure the proportion of infected nodes surrounding it. Thus, the node with the highest label ranking would be regarded as our most vigorous competitors for the propagation source. We implement the basic label ranking algorithm through label propagation to calculate the label ranking scores under the situation of complete observation without knowing the details of infection propagation model.

More severe challenges arise while dealing with the situation with incomplete observation. Only part of the network status is known. Our framework for identifying the propagation source with snapshot is divided into two stages in this paper. The complete network status can be inferred via semi-supervised classification in the first stage. In this way, we can solve the issue with snapshot in the second stage in the same way we do under the situation with complete observation based on the inferred result.

Label ranking algorithms mentioned in this paper are capable of locating the single source with low computational complexity which has an approximate linear relationship with the number of edges. Experiments on several synthetic and real-world datasets prove the validity of proposed methods. Meanwhile, the unified label ranking framework would support more advanced algorithms, such as graph neural networks, to obtain the label ranking scores or restore the network status.

## Materials and methods

### Problem description

At time $t < 0$, state of all nodes in an unweighted, undirected contact network $G(V, E)$ remain susceptible (S). A node $s \in V$ become infected at time $t = 0$, which makes it the single propagation source. The pathogen propagation follows SI/SIR model in [8, 9]. Given a time $t = T$ during the propagation process, observation vector of the network $O$ will provide the exact state (S/I/R) of all nodes or some of them.

The following source identification algorithms aim to locate the single propagation source $s$ on the basis of complete observation or snapshot $O$ without knowing $T$.

### Label ranking algorithm with complete observation

Inspired by source prominence [27] and manifold ranking [29, 30], we attempt to address the single propagation source identification problem based on the idea of *label ranking*.

Firstly, we believe that in addition to the centrality of the source in the infection subgraph, surrounding uninfected nodes should be taken into account as well [12, 26, 27]. Considering the forward process of propagation, pathogens spread from the source $s$ on contact network and individuals surrounding $s$ are then infected hierarchically. It means that the majority of the source's neighboring nodes should be infected, which makes the source in the center of infection subgraph and far from the uninfected frontier as well. This can be explained by

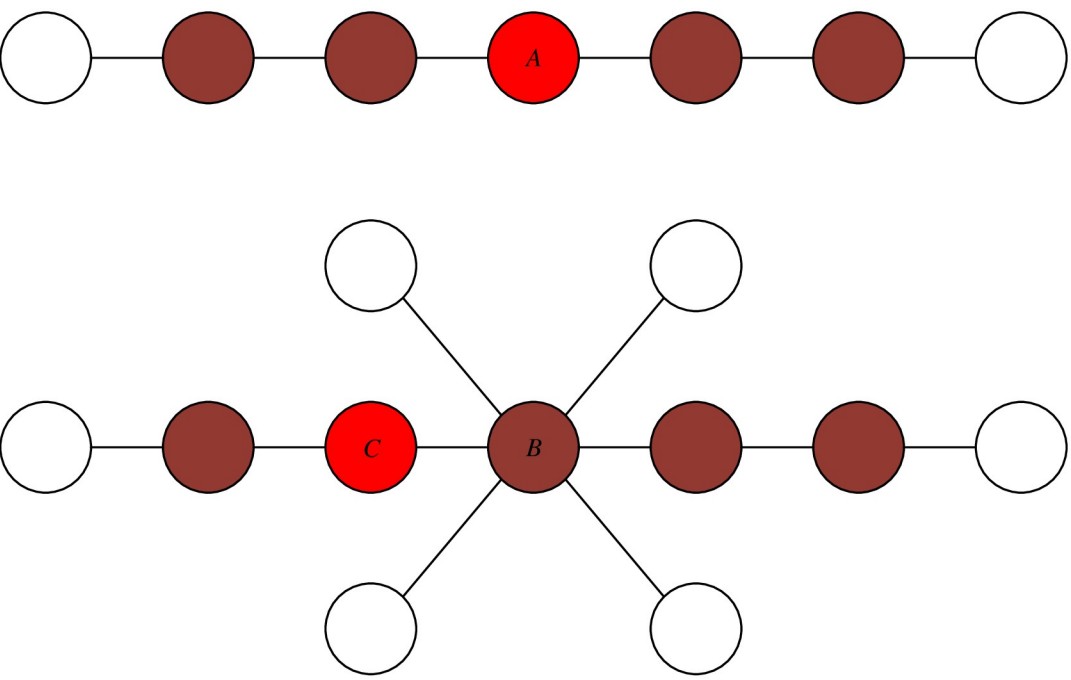

**Fig 1. The illustration of infection subgraph and uninfected frontier.** Uninfected nodes should be considered. Infection snapshot examples (colored nodes are infected while the red node are the true sources) In the first network, node A is the center of infection subgraph and the source as well; In the second network, node B is the center of infection subgraph, but amounts of uninfected nodes reduce its probability to become the source. Node C is the actual source. Modified from [12].

Fig 1 [12]. Therefore, individuals surrounded by more infected nodes and less uninfected nodes would have a higher likelihood of being the propagation source.

Based on the above intuition and properties about the source, we borrow ideas from [26] to find a reasonable label assignment $\boldsymbol{f}$ for the nodes in network $G$ where each label ranking score $f_i$ represents the relative amount of infected and uninfected nodes surrounding the corresponding individual $i$. In this way, we infer that the node with the highest label ranking would be the most possible propagation source. While Google's PageRank algorithm effectively measures the significance of web pages [31], we expect that label ranking on this issue could measure the probability of nodes to become the source.

We design the following Basic Label Ranking for Source Identification algorithm (Alg. 1) to calculate the label assignment of all individuals, which could help us locate the propagation source effectively.

**Algorithm 1** Basic Label Ranking for Source Identification

**Input:** contact network $G(V, E)$, label vector of complete observation $\boldsymbol{y}$, parameter $\alpha$
**Output:** single propagation source $s^*$
1: **function** LABELRANKINGSCORE($G$, $\boldsymbol{y}$, $\alpha$)
2:    initialize $G'$s adjacency matrix $\boldsymbol{W}$
3:    initialize diagonal matrix $\boldsymbol{D}$ with $D_{ii} = \sum_{j=1}^{|V|} W_{ij}$
4:    $\boldsymbol{S} \leftarrow \boldsymbol{D}^{-1/2}\boldsymbol{W}\boldsymbol{D}^{-1/2}$
5:    $\boldsymbol{f}^{(0)} \leftarrow \boldsymbol{y}$
6:    $k \leftarrow 0$
7:    **while** $\boldsymbol{f}^{(k)}$ does not reach the convergence $\boldsymbol{f}^*$ **do**
8:      **for** each node $i$ **do**
9:          $\boldsymbol{f}_i^{(k+1)} \leftarrow \alpha\sum_{j\in N(i)}\boldsymbol{S}_{ij}\boldsymbol{f}_i^{(k)} + (1-\alpha)\boldsymbol{y}_i$

```
10:      end for
11:      k ← k + 1
12:    end while
13:    return f*
14: end function
15: function BLRSI(G, y, α)
16:    f* = LabelRankingScore (G, y, α)
17:    s* = argmax_{i∈V,y_i=1} f_i*
18:    return s*
19: end function
```

Now, we will give the detailed explanations about the steps of this label ranking algorithm.

Firstly, label vector $y$ is applied to express the initial state of the nodes in network $G$. Based on the above-mentioned idea, we would simply be concerned about two categories of nodes, the uninfected ones (state S in SI/SIR model) and the infected ones (state I in SI model, state I/R in SIR model). As individuals surrounded by more infected nodes should be given with higher labels (which means higher probability to become the source), the initial label would be positive for infected ones and negative for uninfected ones. We can thus define the initial label vector $y$ from the complete observation $O$ as follows:

$$y_i = \begin{cases} 1 & O_i = \text{I/R} \\ -1 & O_i = \text{S} \end{cases} \tag{1}$$

Further discussion on the initialization of label vector $y$ will be given later.

The second step is label propagation on the network. We would build the adjacency matrix $W$ of $G$ and its symmetrically normalized form $S$. $S_{ij}$ represents the propagation probability from node $j$ to node $i$. Label vector $f$ is initialized with $y$. During each iteration of the label propagation (Line 9), each node receives labels from its neighbors, and retains its initial state in the meantime. $N(i)$ represents the set of node $i$'s neighbors. The relative amount of the label information from nodes' neighbors and their initial labels is controlled by parameter $\alpha$ ($0 < \alpha < 1$). For a certain node, its infected neighbors would provide it positive labels as incentive and uninfected neighbors would provide negative ones as punishment. The process would go on during the label propagation in order to reflect the impact of distant nodes.

Meanwhile we would give two possible variants of Alg. 1 by modifying the normalized matrix $S$ with the stochastic matrix $P = D^{-1}W$ or its transpose $P^T$ [29].

When the label vector reaches its convergence $f^*$, each element $f_i^*$ would represent the final label ranking score of individual $i$. A node with higher ranking means that it's surrounded by more infected nodes and less uninfected nodes, which makes it more probably to become the source according to our former discussion. Therefore, we would infer the propagation source $s^*$ from the convergent label vector $f^*$ in the third step. The source should have the highest ranking among the nodes that are infected according to the observation $O$. The infected state of candidate nodes ensure that the inferred result could become the source while the highest ranking makes the inferred result most likely to become the source.

## Algorithm analysis of basic label ranking

The above label ranking algorithm aims to minimize the following cost function [29]:

$$\min_f \frac{1}{2} \left( \sum_{i,j=1}^n W_{ij} \left\| \frac{f_i}{\sqrt{D_{ii}}} - \frac{f_j}{\sqrt{D_{jj}}} \right\|^2 + \mu \sum_{i=1}^n \|f_i - y_i\|^2 \right) \tag{2}$$

The cost function can be divided into two terms, which are the smooth constraint and the fitting constraint, respectively.

The smooth constraint in the first term means that the label ranking scores shouldn't change too much between nearby nodes. The degree of each node is applied to get an unbiased label ranking score in this constraint. As a local measurement, the degree of node $i$ represents the number of $i$'s neighbors. Higher degree will result in receiving label values from more nearby nodes. The bias between different nodes caused by the original network topology can be eliminated through dividing label ranking score by the square root of degree $D_{ii}$, similar to the study in [24]. This bias eliminating process ensures that the label ranking scores of nodes with higher degree won't be affected significantly by their low-degree neighbors due to the smooth constraint. In this way, higher label ranking score of a node reasonably indicates that this node tends to be surrounded by more infected individuals, which is consistent with the intention of label ranking. Intuitively, the source is likely to have more infected nodes and less uninfected nodes surrounded. Thus, the node with the highest label ranking will be the most likely one to become the propagation source. The fitting constraint in the second term ensures that the label ranking scores shouldn't deviate much from the initial label vector. The parameter $\mu$ is used to capture the trade-off between these two constraints.

With the above regularization framework, Alg. 1 can obtain the reasonable label ranking and identify the propagation source afterwards.

Based on the iteration equation of BLRSI algorithm above, the convergence result of the label vector is given by [29]:

$$\boldsymbol{f}^* = (1 - \alpha)(\boldsymbol{I} - \alpha\boldsymbol{S})^{-1}\boldsymbol{y} \qquad (3)$$

where $\boldsymbol{I}$ is an $n \times n$ identity matrix ($n = |V|$). Therefore, the label vector $f$ would finally converge through label propagation.

Then we will discuss the time complexity of Alg. 1. The running time of building the matrices $\boldsymbol{W}$, $\boldsymbol{D}$ and $\boldsymbol{S}$ is $O(m)$ ($m = |E|$), as there are $2 \times m$ non-zero elements in all these matrices. For the label propagation step (Lines 8-11), each iteration's running time is $O(m)$ as well. The overall complexity of Alg. 1 is $O(km)$, where $k$ is the number of iterations.

In another way, the convergence result can be calculated directly by Eq (3). However, the time complexity of solving the inverse of $n \times n$ matrix ($\boldsymbol{I} - \alpha\boldsymbol{S}$) would be $O(n^3)$ through Gaussian Elimination. As the purpose of Alg. 1 is to find the node with the highest label ranking, highly strict convergence condition won't be necessary. Thus, the number of iterations $k$ can be controlled in an acceptable range. Combining with the fact that the contact network is always sparse ($m \ll n^2$), we figure out that the time complexity of label propagation is much smaller than solving Eq (3) directly, which makes Alg. 1 be of more practical value.

Apparently, complexity of the two variants is exactly the same as original Alg. 1.

## Another view of label ranking

Label Ranking algorithm Alg. 1 can be analyzed from the view of message passing. Traditional label propagation based methods propagate labels in networks to deal with semi-supervised [29, 32] or unsupervised tasks [33]. Recent years, Graph Neural Networks (GNNs) [34, 35] have been generally studied to apply deep learning on non-Euclidean structures. Among these methods, Message Passing Neural Networks (MPNNs) [36] propose a unified framework for several graph convolution networks. MPNNs consist of two phases, the message passing phase

and the readout phase. The message passing phase includes $T$ graph convolution operations to propagate features or messages in networks. Each operation can be described as:

$$
\begin{aligned}
m_v^t &= \sum_{w \in N_v} W_t(h_v^{t-1}, h_w^{t-1}, e_{vw}) \\
h_v^t &= U_t(h_v^{t-1}, m_v^t)
\end{aligned}
\tag{4}
$$

$h_v^t$ is the hidden state of node $v$ at iteration $t$ and $e_{ew}$ is the edge feature. If we treat the status of nodes in contact networks as the input features, the label propagation process of Alg. 1 (Lines 8-10) can be unified into the message passing phase approximately. The vertex update function $U_t$ would be $(1 - \alpha)h_v^0 + \alpha m_v^t$ while the message function $W_t$ would be $S_{vw} * h_w^{t-1}$ in this case. $T$ is the iteration number of Alg. 1. Since we could set the termination condition, $T$ would be a finite number. The readout function in MPNNs is defined as:

$$
\hat{y} = R(h_v^T | v \in G)
\tag{5}
$$

The label ranking algorithm would simply use an $\mathrm{argmax}(\cdot)$ operation as the readout function to infer the potential propagation source. In this way, our label ranking algorithm can be viewed as a specific implement of MPNNs.

## Label ranking algorithm with snapshot

In this part, we will continue to focus on the source identification problem with snapshot. Firstly, Vanilla Label Ranking Algorithm directly inspired by the study before is provided. A novel solution for source identification with snapshot is then proposed to work out the problem through two stages.

The source identification problem with snapshot differs from the origin problem description in the form of observation $O$. In actual scenes, complete observation of the state of all individuals or all infected ones is too grueling and time-consuming to obtain. Snapshot provides partial information about network status, e.g. the state of a certain node given by snapshot $O$ can be known or unknown. Some researchers may assume that only a portion of the infected nodes is observed [16, 28]. In this way, massive highly accessible information from uninfected nodes is neglected. However, we believe both infected and uninfected nodes make a contribution to locating the propagation source based on our study before and other work [12, 17, 18, 26]. Also, the exact status of uninfected individuals is easy to verify by sampling.

The snapshot can thus described as $O = \{O_L, O_U\}$ where $L$ is the set of observed nodes and $U$ is the set of unobserved ones ($L \cup U = V$). We assume that $|L| = r|V|$. The objective of this problem is to detect the single source $s^*$ given snapshot $O$ without knowing the propagation time $T$.

**Vanilla label ranking algorithm.** We manage to modify the basic idea of label ranking to adapt to the situation of incomplete observation. As is mentioned above, we believe that nodes surrounded by larger proportion of infected nodes would be more likely to become the source. Now that only partial information of the state is given, Vanilla Label Ranking for Source Identification (Alg. 2) is designed to find the target node with the most known infected nodes and

the least known uninfected ones around it. The initial label vector $\boldsymbol{y}$ is defined as:

$$
y_i = \begin{cases} 1 & i \in L, O_i = \mathrm{I/R} \\ -1 & i \in L, O_i = \mathrm{S} \\ 0 & i \in U \end{cases} \tag{6}
$$

**Algorithm 2** Vanilla Label Ranking for Source Identification
**Input:** contact network $G(V, E)$, label vector of snapshot $\boldsymbol{y}$, parameter $\alpha$
**Output:** single propagation source $s^*$
**function** VLRSI $(G, \boldsymbol{y}, \alpha)$
  $f^*$ = LabelRankingScore $(G, \boldsymbol{y}, \alpha)$
  $s^* = \mathrm{argmax}_{i \in V} f_i^*$
  **return** $s^*$
**end function**

Note that the initial label vector $\boldsymbol{y}$ does not reflect the true state of the nodes, we simply choose the node with the highest label ranking as our inference result (Line 3). The convergent proof and complexity of Alg. 2 is identical to Alg. 1.

**Two-stage source identification with snapshot.** On account of our work, the cost function Eq (2) of basic and vanilla label ranking algorithm consists of the smooth constraint and the fitting constraint. The key distinction between these two algorithms is the initial label vector $\boldsymbol{y}$ in the fitting constraint, which is caused by incomplete observation (snapshot). In order to locate the propagation source more precisely, the initial label vector $\boldsymbol{y}$ should approach the actual network status. Thus, we need to estimate each node's state and restore the initial network status from the snapshot.

Based on the above idea, we put forward our Two-Stage Source Identification framework with snapshot. In the first stage, we have to infer the state of each node through the given information from the snapshot to restore the initial network status (complete observation) $\boldsymbol{O}^*$ as accurate as we can. Then the basic label ranking algorithm, or any other source identification method requiring full view of the network can be applied in the second stage to locate the single propagation source. We lay our emphasis on the first stage in this section, as methodologies for the second stage have been widely researched before in this paper and other studies [10].

As we are concerned about only the infected or uninfected state of each individual, the first stage could be transformed into a semi-supervised binary classification issue. The state set would be {I/R, S}, indicating infected and uninfected nodes respectively. Individuals who are observed in snapshot $\boldsymbol{O}$ form labeled data, while those who aren't observed form unlabeled data. We aim to carry out classification on the partially labeled network dataset so that the initial label vector $\boldsymbol{y}$ can be determined accurately.

Semi-supervised learning algorithms have evolved from generative models, self-training and co-training [37] into graph-based [29, 32], manifold-based [38, 39] and deep learning [35, 40] methods in recent years. Among the above-mentioned algorithms, graph-based methods are found out to be appropriate for the classification issue from several aspects. Apparently, the intrinsic graph structure of contact network makes the graph generation procedure in this category of algorithms inessential so that the workload can be significantly reduced. Good interpretation and realizability are their major benefits as well. Thus, we would focus on graph-based methods to restore the network status in this section.

An $n \times 2$ matrix $\boldsymbol{F}$ is defined to correspond to a binary classification on the node set $V$ by assigning the predicted state $O_i^* = \arg\max_{j \leq 2} F_{ij}$ to each node $i$. With the cost function of $\boldsymbol{F}$ and graph structure of network $G$, the semi-supervised classification issue can be worked out

by solving the optimal soft label matrix $F^*$. Interestingly, the graph-based classifier can be restrained under the framework of smoothness constraint and fitting constraint as well [29, 32, 41]. The smoothness constraint ensures that the state of nodes should not change too much between nearby nodes. As a matter of fact, the actual state would only change on the uninfected frontier since the infection subgraph is a connected graph under the circumstance of one single propagation source. The fitting constraint means that the classifier should not deviate too much from the initial state given by the snapshot.

Two classical semi-supervised learning algorithms on the basis of these constraints are then introduced to address the issue of network status restoration. The first algorithm [32] using Gaussian Fields and Harmonic Functions (GFHF) has a cost function in the following form:

$$\min_{F}\infty\sum_{i\in L}\|F_i - Y_{i|L}\|^2 + \frac{1}{2}\sum_{i,j=1}^{n}W_{ij}\|F_i - F_j\|^2 \tag{7}$$

The cost function of the other algorithm [29] using Local and Global Consistency (LGC) is:

$$\min_{F}\frac{1}{2}\left(\mu\sum_{i=1}^{n}\|F_i - Y_i\|^2 + \sum_{i,j=1}^{n}W_{ij}\left\|\frac{F_i}{\sqrt{D_{ii}}} - \frac{F_j}{\sqrt{D_{jj}}}\right\|^2\right) \tag{8}$$

The matrices $W$ and $D$ have exactly the same definitions as those in Alg. 1. $Y = (Y_L, Y_U)$ is the initial state matrix. For $i \in L$, $Y_i$ is a one-hot row vector indicating the node's state corresponding to the state set while it is a zero vector for $i \in U$. Both cost functions consist of the smoothness constraint and the fitting constraint, clearly. The major difference between Eqs (7) and (8) is that the former has an infinite parameter $\infty$ for the fitting constraint to fix the state and soft label of each known node. In this way, the second algorithm LGC would be able to tolerate noise on the snapshot while GFHF would classify the observed nodes as their known state forcibly.

Corresponding optimization methods through label propagation are designed to obtain the optimal classification matrix $F$ as shown in Algs. 3 and 4.

**Algorithm 3** Network Status Restoration via GFHF

```
Input: contact network G(V, E), initial state matrix Y
Output: inferred network status O* indicating nodes' state
 1: function GFHF (G, Y)
 2:    initialize G's adjacency matrix W
 3:    initialize diagonal matrix D with D_ii = ∑_{j=1}^{n} W_ij
 4:    P ← D⁻¹W
 5:    F⁽⁰⁾ = (F_L, F_U) ← Y = (Y_L, 0)
 6:    k ← 0
 7:    while F⁽ᵏ⁾ does not reach the convergence F* do
 8:      for each node i do
 9:          F_i⁽ᵏ⁺¹⁾ ← ∑_{j∈N(i)} P_ij F_i⁽ᵏ⁾
10:      end for
11:      F_L ← Y_L
12:      k ← k + 1
13:    end while
14:    for i = 1 → n do
15:        O_i* = argmax_{j≤2} F_ij*
16:    end for
17:    return O*
18: end function
```

**Algorithm 4** Network Status Restoration via LGC

```
Input: contact network G(V, E), initial state matrix Y parameter α
```

```
Output: inferred network status O* indicating nodes' state
1: function LGC(G, Y, α)
2:    initialize G's adjacency matrix W
3:    initialize diagonal matrix D with Dᵢᵢ = Σⁿⱼ₌₁ Wᵢⱼ
4:    S ← D⁻¹ᐟ²WD⁻¹ᐟ²
5:    F⁽⁰⁾ = (Fₗ, Fᵤ) ← Y = (Yₗ, 0)
6:    k ← 0
7:    while F⁽ᵏ⁾ does not reach the convergence F* do
8:      for each node i do
9:          Fᵢ⁽ᵏ⁺¹⁾ ← α Σⱼ∈N(i) SᵢⱼFᵢ⁽ᵏ⁾ + (1 − α)Yᵢ
10:     end for
11: k ← k + 1
12:   end while
13:   for i = 1 → n do
14:       Oᵢ* = arg maxⱼ≤₂ Fᵢⱼ*
15:   end for
16:   return O*
17: end function
```

The first and the second column of $F$ can be understood as a label that indicates the possibility of being uninfected or infected, respectively. During each iteration of the label propagation process (Alg. 3, Line 9; Alg. 4, Line 9), each node receives labels from its neighbors in two ways based on $P$ or $S$. The distinction of the fitting constraint between GFHF and LGC is reflected somewhat in the algorithms as well. Alg. 3 fixes the soft label of observed nodes in each iteration (Alg. 3, Line 9) while Alg. 4 retains a certain amount of the initial state with parameter $\alpha$ regulating it (Alg. 4, Line 9). After the convergence, larger value of $F_{i1}^*$ than $F_{i2}^*$ means that the node $i$ is similar to the observed infected nodes on manifolds than the uninfected ones. This will make node $i$ more probably to become infected under complete observation, and vice versa. The inferred network status $O^*$ can thus be obtained by $F^*$.

In the second stage, the algorithms designed for complete observation can be applied as the complete network status has been inferred. Here we continue to adopt our Label Ranking Algorithm (Alg. 1) in the second stage. The required label vector $y$ can be initialized by Eq (1) easily. The different algorithms applied in the first stage lead to the two variants of our Two-Stage Source Identification Algorithm with Snapshot as shown in Alg. 5. In this way, we are able to infer the propagation source $s^*$ with incomplete observation of the network in two stages.

**Algorithm 5** Two-Stage Source Identification with Snapshot

```
Input: contact network G(V, E), initial state matrix Y, parameter α₁, α₂
Output: single propagation source s*
1: function TSSI-GFHF(G, Y, α₂)
2:    inferred network status O* ← GFHF(G, Y)
3:    initialize label vector y from O*
4:    s* ← BLRSI(G, y, α₂)
5:    return s*
6: end function
7: function TSSI-LGC(G, Y, α₁, α₂)
8:    inferred network status O* ← LGC(G, Y, α₁)
9:    initialize label vector y from O*
10:   s* ← BLRSI (G, y, α₂)
11:   return s*
12: end function
```

Above all, we propose our Two-Stage Source Identification framework and corresponding algorithms to locate the propagation source with incomplete observation. Classification matrix

*F* and label vector *f* are applied to represent the state probability and the label ranking in stage one and two, respectively. Although *F* and *f* owe different meanings, both of them follow the smooth constraint and the fitting constraint due to the manifold assumption [37] based on the graph structure of contact network *G*.

From the other point of view, the label ranking vector *f* can be seen as a classification function as well. The graph mincut approach [41] defines the objective function with discrete classification vector $f_1 : V \rightarrow \{-1, +1\}$. The original GFHF approach [32] assumes the binary label $y \in \{0, 1\}$ and defines a function $f_2$ together with the harmonic threshold to determine the classification. Though the initial label (state) is known for our label ranking algorithm, vector *f* can represent how strong each node's "infected/uninfected property" is. The higher label ranking score a node has, the more influence the node has from the infected nodes and less from the uninfected ones. That is to say, assuming that its state is unknown, the inferred propagation source node via label ranking would have a higher likelihood of being infected than other nodes. From this point of view, the two stages can be unified into a semi-supervised learning framework from the methodology perspective.

Label propagation is then applied to solve the semi-supervised classification issue in stage one and the label ranking in stage two. The time complexity of both Alg. 3 and Alg. 4 is $O(km)$ analogously, where *k* is the number of iterations in the label propagation process. The overall time complexity of Alg. 5 is $O((k_1 + k_2)m)$. $k_1$ and $k_2$ express the number of iterations in the first and second stage respectively. The two stages can thus be unified into a label propagation framework from the algorithm perspective.

## Dataset description

We test our source identification algorithms on 6 different datasets. The setting of datasets is borrowed from some existing studies [3, 16, 27]. As stated in Table 1, half of them are real-world networks while others are synthetic ones. Since the actual contact networks are complicated and unpredictable, the synthetic networks are used to represent some of the possible properties of networks while real-world ones may reflect the actual situation to some extent. The datasets are described as follows.

Football [42] represents the schedule of Division I games for the 2000 season.

French school [43] is a real-world contact network collected in a French primary school on October 1st, 2009. Edges with contact duration less than 1 minute are removed.

Roget [44] contains cross-references in Roget's Thesaurus, 1879. Maximal connected component is extracted from the original network.

Synthetic networks [45] include three complex network models naming ER, BA and WS, representing random network, scale-free network and small-world network respectively.

**Table 1. Datasets.**

| No. | Dataset | #Nodes | #Edges | #avg(degree) | density | diameter |
|-----|---------|--------|--------|--------------|---------|----------|
| 1 | Football | 115 | 613 | 10.7 | 0.094 | 4 |
| 2 | French school | 236 | 2965 | 25.1 | 0.107 | 5 |
| 3 | Roget | 994 | 3641 | 7.3 | 0.007 | 10 |
| 4 | ER | 200 | 1247 | 12.5 | 0.058 | 4 |
| 5 | BA | 200 | 1164 | 11.6 | 0.060 | 4 |
| 6 | WS | 200 | 1200 | 12.0 | 0.058 | 4 |

We choose the above datasets as the underlying contact network of the propagation process to test the performance of our algorithms. We neglect the potential temporal and weight information to simplify the network topology, making it static, undirected and unweighted.

## Experiment setup

**Propagation model.** We choose discrete SI and SIR model as our propagation model in our experiments. For both SI and SIR model, the transmission rate is set to be 0.3. The cure rate is 0.1 for SIR model. It means that an uninfected node (S) would be infected by any of its infected neighbor with a probability of 0.3 in each time step. The influence of multiple infected neighbors would be calculated independently. An infected node would probably recover weith a probability of 0.1 in each time step as well while we deal with the SIR model. As mentioned above, we concentrate on the infected or uninfected state of individuals. Thus, SIR model can be unified into SI model by treating R state as I state.

**Comparing methods.** To study the effect of label ranking algorithms, we conduct two types of experiments to deal with two kinds of problems: (I) source identification with complete observation and (II) source identification with snapshot.

In (I), we test Alg. 1 (denoted as BLRSI) and its two variants (denoted as BLRSI-P1 and BLRSI-P2 for $P$ and $P^T$ respectively) under SI model with complete observation. We compare our algorithm with Unbiased Betweenness (denoted as UB) in [24], Rumor Centrality (denoted as RC) in [3], single seed algorithm in NetSleuth (denoted as SSNS) [12] and Exoneration & Prominence based Age (denoted as EPA) [26]. These comparing methods are typical heuristic solutions for source identification.

As for (II), most existing proposals for source detection with snapshot consider rather different problem settings with what we mention here. Dynamic Message-Passing [17] requires parameters of propagation model and the upper limit of propagation time to solve the issue. Reverse Infection [16], Reverse Greedy [14] and Score-based Reverse Propagation [28] assume that only part of infected nodes could be observed while no exact information could be given for any uninfected nodes. Since semi-supervised classification in two-stage label ranking algorithms would not work with no labeled infected node, we compare Alg. 2 (denoted as VLRSI) with Reverse Greedy (denoted as RG) under the SIR model firstly. Then we compare our designed algorithms for snapshots including Alg. 2 (denoted as VLRSI) and Alg. 5 (denoted as TSSI-GFHF and TSSI-LGC) to verify the superiority of this two-stage framework.

The label propagation parameter $\alpha$ is set to 0.5 in Alg. 1 and Alg. 2 while $\alpha_1 = \alpha_2 = 0.5$ in Alg. 5. We let other comparing methods use their respective optimal parameters. Due to different numbers of nodes, the convergence conditions for the above algorithms become:

$$\|f^{(k+1)} - f^{(k)}\|_2 \leq 10^{-4}n \tag{9}$$

or

$$\|F^{(k+1)} - F^{(k)}\|_F \leq 10^{-4}n \tag{10}$$

where $n = |V|$ for each network.

All simulations and algorithms are implemented in Python 3.6.

**Evaluation metrics.** Two representative evaluation metrics are firstly introduced to evaluate the performance of single source identification in both (I) and (II). Detection Rate (DR) refers to the probability to detect the propagation source accurately. Average Error Distance (AED) refers to the average geodesic distance between the inferred source and the actual one.

In (II), F1-score is applied to measure the effect of network status restoration in stage one, which is a binary classification issue.

Meanwhile, we will show average run time of each algorithm to depict its time complexity quantitatively.

**Experimental settings.** For both (I) and (II), A node is chosen uniformly in the network to initiate a propagation process in each run. We then run each simulation till at least 30% of the individuals are infected. We use the simulation result as complete observation for (I) directly. As for (II), the snapshot could be generated by SIR model while only infected nodes could be observed to compare VLRSI with RG. For wider application, a proportional of nodes could selected uniformly as a snapshot for source identification. We make sure that each snapshot will contain nodes in both infected and uninfected state in this situation. All reported results are averaged over 1000 independent runs.

## Results and discussion

### Source identification with complete observation

We compare the source identification performance of different algorithms under the situation of complete observation in problem (I). DR and AED are applied to measure the effectiveness of the methods while average run time is applied to measure the efficiency.

The main experimental results are summarized in Table 2. Taken together, we hold the opinion that our proposed algorithm BLRSI outperforms other comparing methods significantly.

Firstly, we will compare our original BLRSI in Alg. 1 with the comparing methods. Average error distance of BLRSI is the smallest among networks 2, 3, 4, 6. This superiority becomes more remarkable on larger networks. We note that EPA might be a better algorithm on networks 1 and 5. EPA achieve a higher DR on network 2 and provides a stiff competition to our proposed algorithms on network 4 as well. However, its unsatisfactory performances on larger and more complicated networks such as Roget restrict its application. There would be a high probability for EPA to identify a node two or more hops away from the actual source as its inferred result incorrectly on these networks. The higher stability of BLRSI in both large and

**Table 2. Source identification performance with complete observation.**

| network | 1.Football | | | 2.French School | | | 3.Roget | | |
|---|---|---|---|---|---|---|---|---|---|
| algorithm | DR | AED(hop) | time(ms) | DR | AED(hop) | time(ms) | DR | AED(hop) | time(s) |
| UB | 0.067 | 1.684 | 13.612 | 0.035 | 1.662 | 110.576 | 0.030 | 2.219 | 0.955 |
| RC | 0.246 | 1.118 | 156.252 | 0.066 | 1.506 | 624.445 | 0.055 | 2.487 | 15.639 |
| SSNS | 0.038 | 1.838 | **2.049** | 0.011 | 1.878 | **20.974** | 0.013 | 3.196 | **0.241** |
| EPA | **0.399** | **0.747** | 24.854 | **0.130** | 1.473 | 385.10 | 0.039 | 3.422 | 1.949 |
| BLRSI | 0.231 | 0.937 | 7.779 | 0.078 | **1.123** | 39.202 | 0.059 | **1.759** | 0.573 |
| BLRSI-P1 | 0.230 | 0.968 | 8.007 | 0.049 | 1.437 | 39.677 | **0.116** | 2.415 | 0.497 |
| BLRSI-P2 | 0.212 | 0.998 | 8.118 | 0.053 | 1.175 | 39.822 | 0.046 | 1.994 | 0.631 |
| network | 4.ER | | | 5.BA | | | 6.WS | | |
| algorithm | DR | AED(hop) | time(ms) | DR | AED(hop) | time(ms) | DR | AED(hop) | time(ms) |
| UB | 0.109 | 1.445 | 50.605 | 0.020 | 1.542 | 60.698 | 0.083 | 1.676 | 54.826 |
| RC | 0.241 | 1.246 | 481.517 | 0.027 | 1.527 | 399.900 | **0.244** | 1.272 | 598.586 |
| SSNS | 0.029 | 1.942 | **9.909** | 0.006 | 2.103 | **11.723** | 0.034 | 1.855 | **10.368** |
| EPA | 0.260 | 1.308 | 99.229 | **0.343** | **0.970** | 91.170 | 0.187 | 1.457 | 108.853 |
| BLRSI | **0.262** | **1.078** | 20.448 | 0.118 | 1.236 | 18.574 | 0.220 | **0.982** | 20.334 |
| BLRSI-P1 | 0.257 | 1.215 | 21.104 | 0.189 | 1.553 | 19.397 | 0.236 | 1.030 | 21.081 |
| BLRSI-P2 | 0.188 | 1.201 | 20.771 | 0.088 | 1.287 | 19.565 | 0.203 | 1.051 | 20.759 |

small networks makes it a practical source identification algorithm. We will show the distributions of error distances for further discussion in S1 Fig. We can see that the sources estimated by BLRSI is close to the true sources. Even if BLRSI cannot identify the propagation source $s$ accurately, the inferred source $s^*$ is usually one or two hops away from it. From another aspect, we observe that the detection rates in this source identification issue tend to be low (no more than 40%). This will result in contingency and volatility of detection rate because of finite experiments. So we believe that error distance and its distribution will be more valuable to measure the validity and practicability of the methods. Higher AED could lead to more bad cases in real scenarios, which administrators would not want to see. In this way, the superiority of BLRSI is further verified based on our discussion.

Then we will focus on the efficiency of source identification algorithms mentioned above. Time complexity of these algorithms is shown in Table 3. The time complexity dominates the speed of detecting the propagation source generally. As SSNS has a linear relation with the number of edges $m$, it has the least run time on our datasets. However, BLRSI, which is also approximate linear with $m$, is superior to SSNS prominently due to its higher detection rate and smaller error distance. To be specific, the BLRSI only take 4 or 5 iterations to achieve convergence (9) or (10) in our experimentations. Hence, the time cost of BLRSI is basically in the same magnitude with SSNS, which could be verified in Table 2. The other three algorithms UB, RC and EPA with time complexity $O(n^2) \sim O(n^3)$ have better performance than SSNS. However, their run time is much longer than BLRSI.

Among the existing source identification algorithms, those with decent performance are usually computationally expensive with time complexity of $O(tmn)$ [17], $O(n^3)$ [14, 16] or higher, which will limit their practical value severely. Considering our evaluation metrics comprehensively however, we find out that our proposed method BLRSI is an outstanding algorithm for source identification, which could take account of both effectiveness and efficiency. The BLRSI method will have important application value to locate the propagation source accurately within acceptable time even on large scale networks.

We compare the two variants of BLRSI with the original one as well. Due to the results in Table 2, the original BLRSI outperforms the other ones from the view of average error distance. It makes Alg. 1 the most excellent one among the source identification methods we study in this section. The performance of BLRSI-P2 with probability transition matrix $P^T$ is close to BLRSI in general. Though BLRSI-P1 with matrix $P$ replacing $S$ has better detection rates on certain datasets, long error distances and instable performance make it lack of application value. Based on this, we can conclude that the bias elimination in Eq (2) does make a difference in source identification.

In another way, the results in Table 2 would reveal the relationship between network structures and source identification metrics. Network scales would affect the performance of DR and AED obviously. It would always be harder to locate a single source on a greater network. Meanwhile, algorithms would achieve worse AED on networks with larger diameters and lower average degrees such as Roget, since the distances between nodes would be far away.

**Table 3. Summary of time complexity of source identification algorithms.**

| Algorithm | Time Complexity |
|:---:|:---:|
| UB | $O(n^2)$ |
| RC | $O(n^2)$ |
| SSNS | $O(m)$ |
| EPA | $O(n^2) \sim O(n^3)$ |
| BLRSI | $O(km)$ |

**Table 4. Source identification performance with snapshot under SIR model.**

| network | 2.French school | | | 3.Roget | | |
|---|---|---|---|---|---|---|
| algorithm | DR | AED(hop) | time(ms) | DR | AED(hop) | time(ms) |
| RG | **0.055** | 2.103 | 62.015 | 0.010 | 3.142 | 438.587 |
| VLRSI | 0.040 | **1.260** | **32.973** | **0.016** | **2.345** | **244.712** |
| network | 5.BA | | | 6.WS | | |
| algorithm | DR | AED(hop) | time(ms) | DR | AED(hop) | time(ms) |
| RG | 0.018 | 2.122 | 20.831 | 0.002 | 2.828 | 25.866 |
| VLRSI | **0.019** | **1.615** | **12.479** | **0.122** | **1.304** | **13.783** |

While dealing with networks with higher densities and smaller diameters, source identification algorithms could easily achieve better AED metrics. Among the three synthetic networks, infection source on ER network is the easiest to identify for our source identification algorithms.

## Source identification with snapshot

Firstly, we compare VLRSI (Algorithm 2) with RG under the SIR model. We choose network 2, 3, 5 and 6 as our testing datasets. Both DR and AED are considered as our metric. The results are summarized in Table 4.

We can see that, the proposed VLRSI could achieve pretty results even without exact information about uninfected nodes. Though RG could achieve a better DR on Roget, the lower AED metrics ensure the stability of this vanilla label ranking algorithm.

We adopt our proposed methods under incomplete observation to deal with the source identification issue with snapshot further. Different proportions $r$ of all nodes in contact network $G$ are used to generate the snapshot. In this way, we can evaluate the performance of the proposed algorithms under all kinds of situation. Except for the metrics mentioned before, F1-score is used to measure the performance of the first stage for our two-stage algorithms TSSI-GFHF and TSSI-LGC.

The performance of our proposed source identification algorithms is shown in Fig 2. AED is chosen as the main evaluation metric. We observe that all three label ranking algorithms can

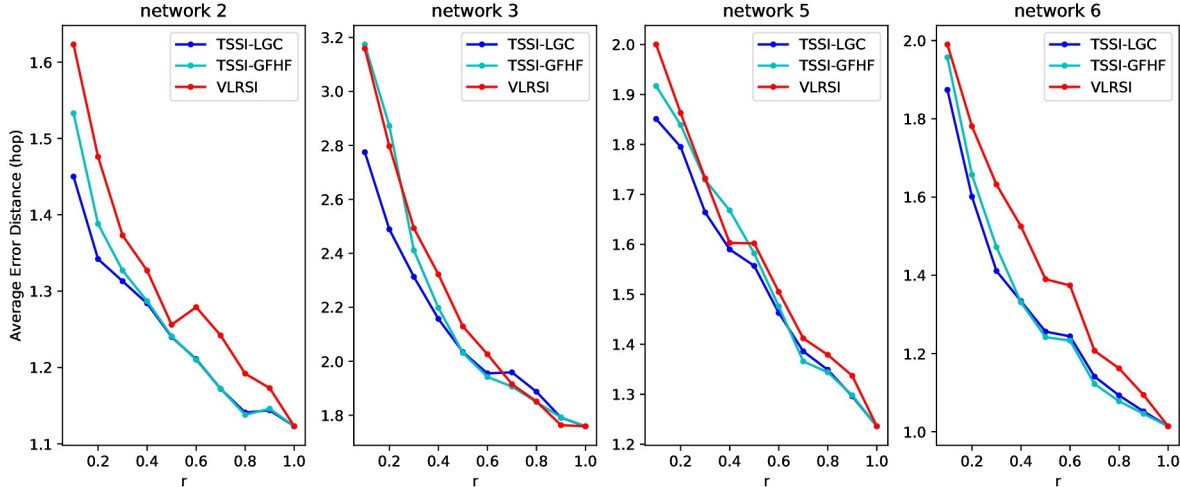

**Fig 2. Average error distance of source identification with snapshot using TSSI-LGC, TSSI-GFHF and VLRSI on network 2, 3, 5, 6.**

effectively locate the propagation source while different values of *r* are given. VLRSI turns out to be the worst among them, which makes our two-stage framework be meaningful. The major difference between the complexity of VLRSI and TSSI is the number of iterations. With an acceptable time cost in the first stage, TSSI can improve the performance of source identification with effect, especially when *r* is small.

Comparing two kinds of TSSI, TSSI-LGC outperforms TSSI-GFHF generally on the four datasets. This can be explained by F1-score in the first stage in Fig 3. LGC can infer the state of unknown nodes more accurately than GFHF. The superiority becomes remarkable with smaller snapshots. It means that fixing the state of known nodes is not necessary for network status restoration. TSSI-LGC is an outstanding source identification method on the whole. Even when only about 50% of nodes in the network can be observed, TSSI-LGC is able to achieve a close result to the comparing methods we study before with complete observation. Thus we are able to draw the conclusion that our two-stage framework for source identification with snapshot can be effective.

## Impact of parameters

**Label propagation parameter $\alpha$.** The parameter $\alpha$ in Alg. 1 is introduced to control the balance between the fitting and smooth constraint. Here we investigate the impact of $\alpha$ in problem (I) via analyzing the change of average error distance caused by different values of $\alpha$.

The values of $\alpha$ range from 0.1 to 0.9 to capture the influence of the two constraints. Fig 4 shows the average error distances for source identification on network 1, 2, 5 and 6. We observe that when $\alpha$ approaches 0 or 1, average error distance of basic label ranking algorithm will increase. The best performance can be obtained when $\alpha$ is around 0.5. Thus we conclude that both the fitting constraint and the smooth constraint play significant role in determining the label ranking. A moderate value of $\alpha$ should be selected in order to identify the propagation source accurately.

**Initial label vector $y$.** The initial label vector $y$ is applied to measure the importance of infected and uninfected state. We denote the original label vector in Eq (1) as $(-1, +1)$, which means $-1$ for uninfected nodes and 1 for infected nodes. In this way, the same weight value is given to uninfected and infected nodes while the sign stands for punishment and incentive respectively.

We point out that the propagation source would be the one surrounded by more infected nodes and less uninfected nodes. If we believe that being far away from the infected region is more important than being surrounded by masses of infected nodes, we can modify the initial label vector $y$ to $(-2, +1)$, for example, so that higher weight value is given to uninfected nodes to reflect harsher punishment.

We conduct an experiment on datasets 2, 3, 5, 6 to show the impact of different initial label vectors in problem (I) briefly. Table 5 summarizes our experimental results. Experimental findings differ on different networks. Since omitting the impact of either infected nodes or uninfected ones yields poor results in all experiments, we would verify our intuition which points out that both infected and uninfected nodes would play roles for source identification. Retaining the original assignment in Eq (1) could reduce the error distance of our inference by BLRSI on networks 2, 5, 6. Meanwhile, higher DR on network 5 and lower AED on network 3 could be achieved by giving higher weight value to uninfected nodes. One reasonable suspicion is that specific network topologies and scales lead to the divergence of our conclusions. The appropriate assignment of initial label vector could be explored further.

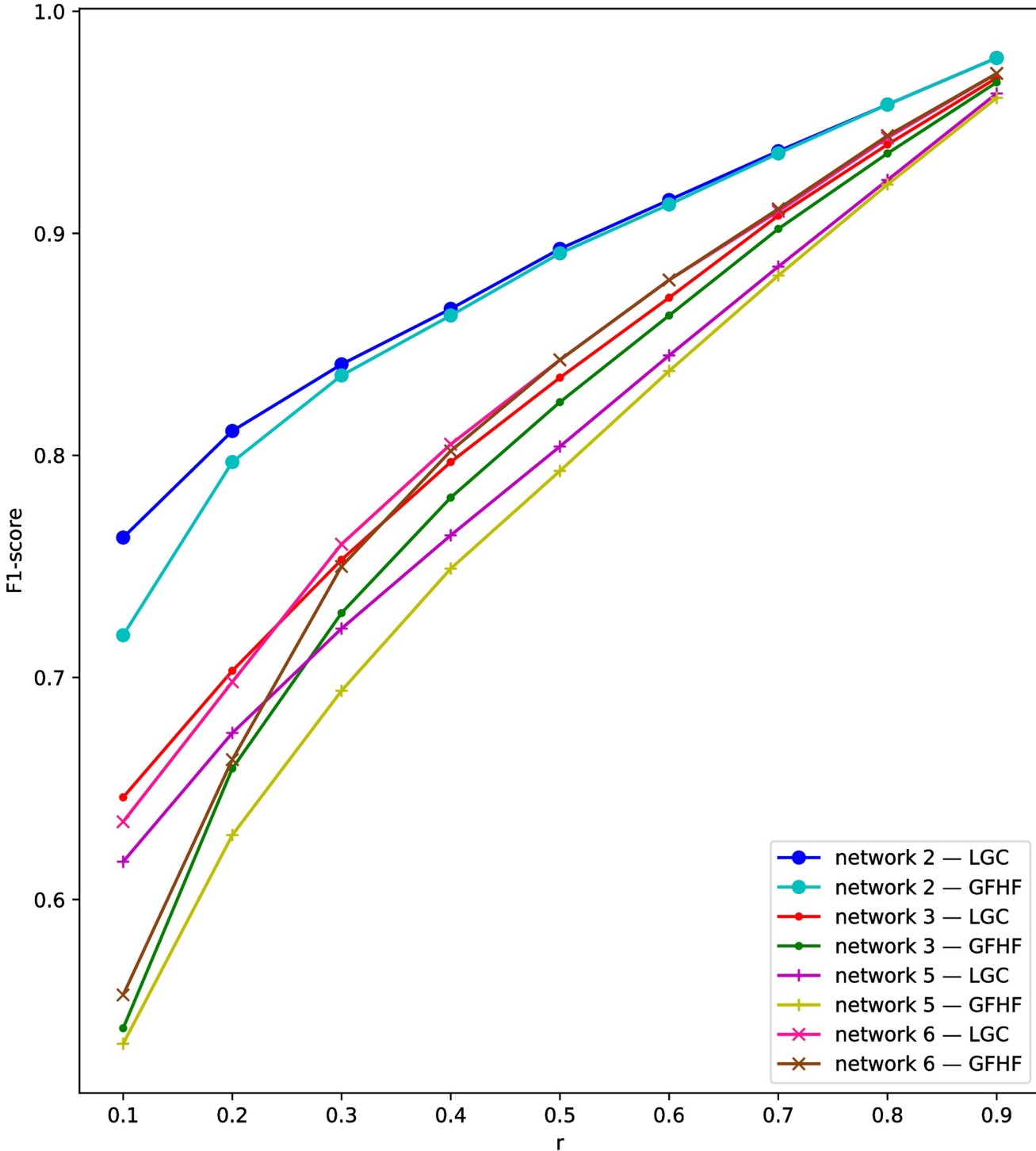

**Fig 3. F1-scores of network status restoration in the first stage using semi-supervised learning on network 2, 3, 5, 6.**

## Implications

Inspired by graph-based semi-supervised learning, we propose a series of label ranking algorithms to infer the propagation source of infectious disease. Compared with existing research, the proposed algorithms have standout performances in three ways.

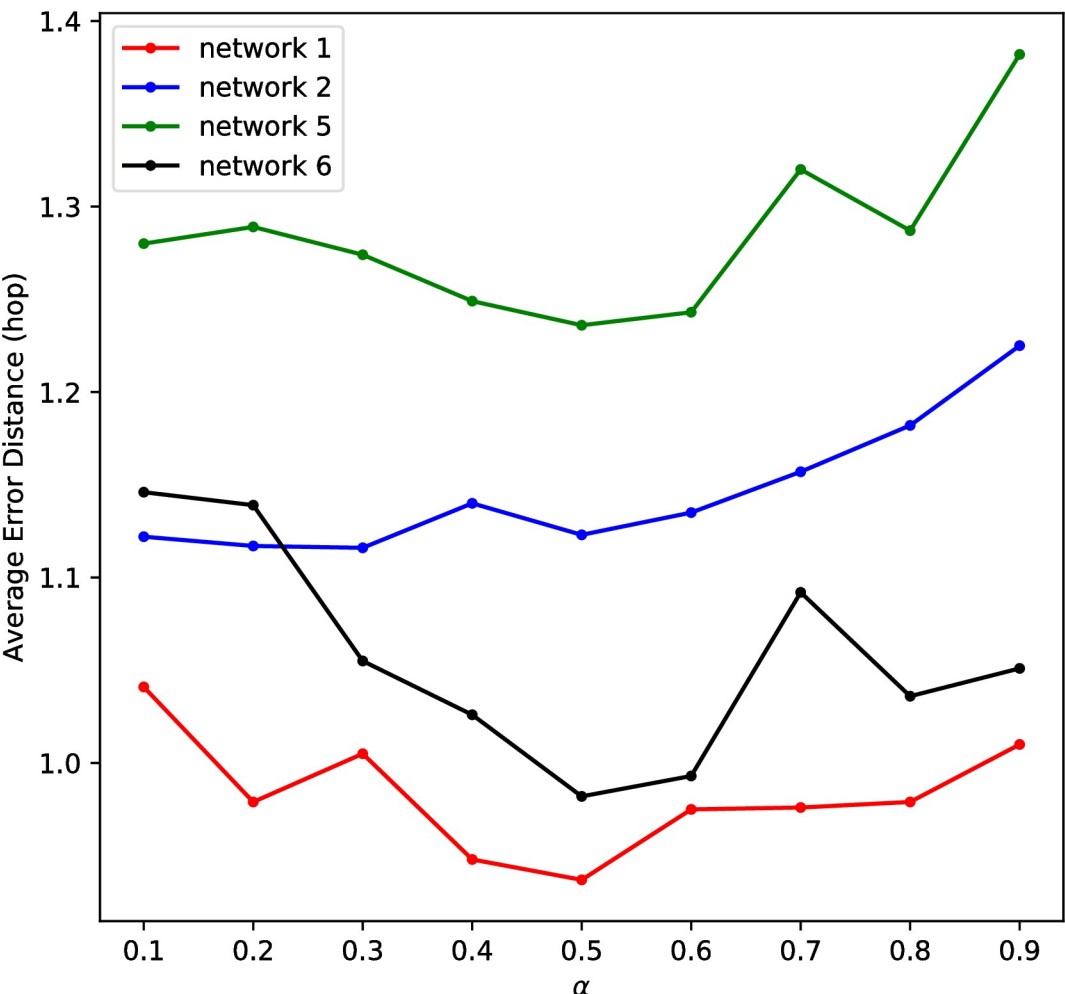

**Fig 4. Impact of parameter $\alpha$ while applying BLRSI on network 1, 2, 5, 6.**

In the first place, the label ranking methods are verified to have higher detection rate and lower error distances through our experiments. From the distributions of error distances on different networks, it can be seen that most source candidates inferred by label ranking would be 0 or 1 hop away from the actual sources. Though we couldn't locate the source all the time, low error distances of label ranking algorithms would provide an important reference for disease screening and source detecting in practice.

**Table 5. Impact of initial label vector.**

| network | 2.French School | | 3.Roget | | 5.BA | | 6.WS | |
|---------|-----------------|----------|---------|----------|------|----------|------|----------|
| $\gamma$ | DR | AED(hop) | DR | AED(hop) | DR | AED(hop) | DR | AED(hop) |
| $(-1, 1)$ | **0.078** | **1.123** | 0.068 | 1.828 | 0.118 | **1.236** | **0.220** | **0.982** |
| $(-1, 2)$ | 0.067 | 1.149 | 0.030 | 1.960 | 0.050 | 1.321 | 0.180 | 1.048 |
| $(-2, 1)$ | 0.049 | 1.124 | 0.076 | **1.646** | **0.174** | 1.322 | 0.205 | 1.028 |
| $(0, 1)$ | 0.060 | 1.187 | 0.024 | 2.148 | 0.022 | 1.510 | 0.140 | 1.138 |
| $(-1, 0)$ | 0.052 | 1.531 | **0.082** | 2.890 | 0.148 | 1.693 | 0.213 | 1.101 |

The time complexity of these proposed algorithms are applicable to large-scale real-world contact networks meanwhile. We could be able to infer the possible propagation source within computational complexity which has an approximate linear relationship with the number of network edges. While dealing with networks of thousands, millions or billions of nodes, the rapidity of label ranking algorithms would be conductive to locating the possible sources for decision makers.

Extendibility of this label ranking framework and the specific methods are worth mentioning as well. Since the label ranking algorithms are derived from the graph structure of contact networks, both the application scenarios and the network models can be extended. Propagation in other networks that can be described by graph including computer networks, social networks and power networks could provide broad application space for label ranking algorithms. Further, more complicated network models which contain weights, edge or vertex attributes and temporal information can be studied based our findings. Moreover, the label ranking framework would support advanced algorithms, including the graph neural networks we mentioned before, to calculate label ranking scores for nodes or restore the network status.

## Limitations

Source identification in networks has been widely studied nowadays while its application in real-world hasn't stepped up yet. The infection source localization of infectious diseases still relies on inspection and medical knowledge. This paper is not immune to lack of practicality either. Two main factors are believed to be behind this phenomenon.

Firstly, the abstraction of network structure does simplify the modeling, algorithm designing and calculating, it would sacrifice much practical information as well. For instance, temporal information of contacts would restrict the direction and timing of propagation.

Meanwhile, collection of data and verification of methods could be quite difficult. Fine-grained real-world contact networks between people can be obtained through techniques including WiFi and RFID. However, relating propagation data could only be collected on a larger scale like regions or cities. Thus, existing research including ours would apply their methods on real-world networks together with simulated propagation process.

The lack of interpretation would be the main limitation while getting back to our study itself. We attempt to locate the propagation source heuristically. Different label propagation based semi-supervised algorithms are thus applied to find the optimal solution for the heuristic hypothesis. Either the hypothesis itself or the algorithms can be explained in other ways. The parameter selection and necessity of the proposed label ranking methods would hardly be interpreted though we have achieved outstanding results. This may limit the future works of this paper.

## Conclusions

In this paper, we introduce our novel solutions via label ranking framework for source identification under different situations. Given complete observation of the network, our basic label ranking algorithm is designed to find out the top-ranked node with the highest proportion of infected nodes surrounding it, which makes it likely to be in the center of infection subgraph and far from the uninfected frontier. We infer this node as the propagation source heuristically. Both infected and uninfected nodes are included in the algorithm to make full use of the network status. The complexity can be reduced to a low level through label propagation iterations, which is quite remarkable among the related methods.

A two-stage algorithm based on semi-supervised learning and label ranking is further proposed for source identification issue with snapshot by restoring the network status and

locating the source. Numerical results on real-world and synthetic network datasets show that our algorithms compare favorably to existing methods on both effectiveness and efficiency. Generally, low computational complexity and outstanding performance make our label ranking algorithms be of high practical value in propagation source identification.

Since we derive our algorithms directly from the graph structure of contact networks, label ranking can thus be extended to more complex network models such as weighted or directed networks. Recent graph-based semi-supervised learning algorithms including graph neural networks [35] and graph embedding [46, 47] can be further studied to apply on the network status restoration stage in our proposed framework. Since we have unified our label ranking algorithms into the framework of MPNNs, more graph neural network structures could be designed to handle the source identification problems. Most existing source identification methods including ours are verified on datasets with determined structures and finite nodes. To adapt to large-scale real-world networks, especially physical contact networks, estimating the infected areas and inferring fine-grained structures are severe challenges remaining to be studied.

## Supporting information

**S1 Fig. Distributions of error distances of source identification algorithms with complete observation on networks 1-6.** Figures (a)-(f) refer to the distributions of error distances on networks 1-6, respectively.
(TIF)

**S1 File.**
(ZIP)

## Acknowledgments

We wish to acknowledge Liang Li for helpful discussions on previous versions of the paper. We would also like to thank the two anonymous reviewers for their comments, criticism, and recommendations, which enabled a significant improvement of the manuscript.

## Author Contributions

**Conceptualization:** Yuewen Jiang.

**Funding acquisition:** Biqing Huang.

**Investigation:** Yuewen Jiang.

**Methodology:** Jianye Zhou.

**Resources:** Biqing Huang.

**Supervision:** Biqing Huang.

**Validation:** Jianye Zhou, Yuewen Jiang.

**Writing – original draft:** Jianye Zhou.

**Writing – review & editing:** Yuewen Jiang, Biqing Huang.

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
