## [Decision Letter · Decision Letter 0]

30 Jul 2020

PONE-D-20-07163

Source Identification of Infectious Diseases in Networks via Label Ranking

PLOS ONE

Dear Dr. Huang,

Thank you for submitting your manuscript to PLOS ONE. After careful consideration, we feel that it has merit but does not fully meet PLOS ONE’s publication criteria as it currently stands. Therefore, we invite you to submit a revised version of the manuscript that addresses the points raised during the review process.

We look forward to receiving your revised manuscript.

Kind regards,

M. Sohel Rahman, Ph.D.

Academic Editor

PLOS ONE

Journal Requirements:

Reviewers' comments:

Reviewer's Responses to Questions

**Comments to the Author**

1. Is the manuscript technically sound, and do the data support the conclusions?

Reviewer #1: Yes

2. Has the statistical analysis been performed appropriately and rigorously? 

Reviewer #1: Yes

3. Have the authors made all data underlying the findings in their manuscript fully available?

Reviewer #1: Yes

4. Is the manuscript presented in an intelligible fashion and written in standard English?

Reviewer #1: Yes

5. Review Comments to the Author

Reviewer #1: This paper “Source Identification of Infectious Diseases in Networks via Label Ranking” deals with estimating single source of infection in a network. The authors provide approaches for both the scenarios when we have complete and partial information of the nodes. For the former, they propose a label ranking based approach and for the latter, besides a direct approach, they propose a two-stage approach to estimate a source. The first stage deals with network status restoration using semi-supervised learning and the second stage in this two-stage framework is similar to the method which deals with complete information. The proposed methods seem to work well in estimating single source of infection on various synthetic and real-world datasets.

The paper is well written and organized. The main contribution of this paper is the idea of network status restoration (first stage in the snapshot observation case) and the provided techniques, i.e., estimating the actual state of each node in a snapshot graph where we have a partial observation, and restoring the initial graph status. However, I have a few comments which, upon getting properly addressed, I believe, the paper could be considered for publication.

Comment 1. About Lines 36 to 40, Lines 54 to 57, Lines 84-94 and Lines 213 to 215: There is a recent paper (Ali, S. S., et.al., EPA: Exoneration and Prominence based Age for Infection Source Identification. CIKM, Nov. 2019) which exploits and shows the importance of uninfected nodes in source identification. As, discussed in this manuscript, the paper (Ali, S.S., et al., CIKM, 2019) also talks about the prominence of a source, where a source is said to have more infected neighbor concentration than a fringe or the farthest away (from the center of an infection) node. I believe that should be added as a reference in the manuscript, besides the already included manuscript reference numbers [12] (Prakash, Aditya B., et al., ICDM, 2012) and [25] (Wang, Zheng, et al., AAAI, 2017).

Comment 2. About Lines 35 to 36: While this generally holds good, there is no mathematical proof for this. In fact, a few works show that graph centrality measures do a good job in source estimation in certain scenarios. Therefore, to inform the readers of such scenarios, besides what has already been discussed in this manuscript, I would suggest the authors add these two papers as references for the sake of unbiased dissemination of information:

1. Comin, C. H., & da Fontoura Costa, L. (2011). Identifying the starting point of a spreading process in complex networks. Physical Review E, 84(5), 056105.

2. Ali, S. S., Anwar, T., & Rizvi, S. A. M. (2020). A Revisit to the Infection Source Identification Problem under Classical Graph Centrality Measures. Online Social Networks and Media, 100061.

Comment 3. While, I don’t doubt the intentions of the authors, however, if I am not wrong, Algorithm 1 in the manuscript, i.e., BLRSI, seems starkly similar to Algorithm 1 (LPSI) of manuscript reference number [25] (Wang, Zheng, et al., AAAI, 2017). The only difference which I see is that LPSI estimates multiple sources by checking whether the label propagation score of a node with respect to its neighbors is the highest and considers it as one of the sources if that is the case. In the algorithm (BLRSI), presented in this manuscript, the highest label score amongst all the observed infected nodes is considered to be the source. I believe this has already been achieved in (Ali, S.S., et. al., A Revisit to the Infection Source Identification Problem under Classical Graph Centrality Measures. OSNEM, 2020), where, for the purpose of comparison, the authors have tuned the same algorithm (LPSI) to capture a single source in, I believe, exactly the same way as the BLRSI algorithm presented in the manuscript. Therefore, my question is what makes BLRSI algorithm different to those algorithms given BLRSI has been proposed/designed (observe the language in the Abstract Section and Line 102) in this manuscript? (Also, the analysis of BLRSI provided in Section “Algorithm analysis of Basic Label Ranking” is quite similar to the analysis of LPSI in Wang, Zheng, et al., AAAI, 2017.

Comment 4. For complete observation, while the comparison of BLRSI against NetSleuth/SSNS (manuscript ref. no. [12] (Prakash, Aditya B., et al., ICDM, 2012)) is fine, I suggest the comparison should be made against EPA as well (Ali, S.S., et al., CIKM, 2019), given EPA also exploits uninfected nodes to estimate the source of infection. Besides, EPA is a recent work which has been shown to beat NetSleuth/SSNS (Prakash, Aditya B., et al., ICDM, 2012) in single source infection identification scenario.

Comment 5. For snapshot observation, all the three proposed methods, VLRSI, TSSI-GFHF and TSSI-LGC should and can (if I am not wrong) at least be compared with one of the recent existing works. While the authors, from Lines 368-373, do argue that the existing works have different approaches in comparison to their own methods and thus the comparison is not suitable, however, in the reasons specified by them, the authors do not consider the methods (for comparison) which plainly consider the snapshot network to estimate the source from. The direct and simple way is to compare the proposed methods (especially VLRSI) with those methods which work on SIR model, for example with Reverse Infection (RI) method (Zhu et. al., Information source identification in SIR model: A sample-path-based approach, IEEE/ACM Trans. on Networking, 2016). In case, this still is not possible, the authors should explain the same.

Comment 6. I observe the average degree of the datasets used for experimentation is quite high and, hence, the diameter of these graphs would be on the smaller side and density on the higher side. Therefore, the AED might tend to appear on the good side of things. Lower average degree would mean, higher AED. This is also confirmed by the results (Table 2) achieved on Roget dataset with comparatively lower average degree tending to make AED higher. Typically, if the density of the graph is high, AED becomes small (which makes sense, since the diameter is small in the first place) and DR is not so good. This also explains why DR is so low even on ER graph (26%), when on ER graph source identification performance is generally better. I would suggest authors add density values and diameter values of each dataset in the dataset table (Table 1) and explain this relation (between graph diameter/density and AED/DR).

Comment 7. Generally, in how many iterations does BLRSI achieve convergence in the experimentations that authors have conducted? If there is some iteration number, the authors should add that in the manuscript to help future researchers work with an iterative process instead of a convergent one, wherever suitable.

Comment 8. It appears that the methods proposed in this work can estimate an infection source without having any information on the underlying model of infection propagation. Given the importance of this, I suggest the authors mention this clearly in the manuscript.

6. PLOS authors have the option to publish the peer review history of their article (what does this mean?). If published, this will include your full peer review and any attached files.

Reviewer #1: No

---

## [Author Response · Author response to Decision Letter 0]

4 Sep 2020

Dear Reviewers,

We wish to thank you for the time and effort you have spent reviewing our paper "Source Identification of Infectious Diseases in Networks via Label Ranking". We are pleased to note that you have given positive and constructive feedbacks to help us improve our work.

Motivated by your comments, we have reconsidered the details of our work and tried to improve our work. The revised manuscript has been improved significantly as follows:

1. The innovations of our work has been emphasized in the manuscript. We apply semi-supervised learning algorithms to form a label ranking framework for both complete observation and snapshot scenarios. The experimental results prove the effectiveness and efficiency of our framework.

2. Additional comparing methods are added to prove the effectiveness and efficiency of our proposed label ranking algorithms.

 3. More related references and detailed discussions have been complemented.

Our itemized response to your comments, questions and suggestions (repeated below for your convenience) is as follows. 

Reviewer #1:

Comment 1. About Lines 36 to 40, Lines 54 to 57, Lines 84-94 and Lines 213 to 215: There is a recent paper (Ali, S. S., et.al., EPA: Exoneration and Prominence based Age for Infection Source Identification. CIKM, Nov. 2019) which exploits and shows the importance of uninfected nodes in source identification. As, discussed in this manuscript, the paper (Ali, S.S., et al., CIKM, 2019) also talks about the prominence of a source, where a source is said to have more infected neighbor concentration than a fringe or the farthest away (from the center of an infection) node. I believe that should be added as a reference in the manuscript, besides the already included manuscript reference numbers [12] (Prakash, Aditya B., et al., ICDM, 2012) and [25] (Wang, Zheng, et al., AAAI, 2017).

Response 1. We have found out that the paper (Ali, S.S., et al., CIKM, 2019) would be a good complement for our work. Authors gave the theoretical explanation of the importance of uninfected nodes in source identification tasks, which supports our BLRSI algorithm. We add this paper as a reference in the manuscript. 

Comment 2. About Lines 35 to 36: While this generally holds good, there is no mathematical proof for this. In fact, a few works show that graph centrality measures do a good job in source estimation in certain scenarios. Therefore, to inform the readers of such scenarios, besides what has already been discussed in this manuscript, I would suggest the authors add these two papers as references for the sake of unbiased dissemination of information:

1. Comin, C. H., & da Fontoura Costa, L. (2011). Identifying the starting point of a spreading process in complex networks. Physical Review E, 84(5), 056105.

2. Ali, S. S., Anwar, T., & Rizvi, S. A. M. (2020). A Revisit to the Infection Source Identification Problem under Classical Graph Centrality Measures. Online Social Networks and Media, 100061.

Response 2. We have mentioned (Comin, C.H., et al. Phys Rev E, 2011) as a typical source identification algorithm based on graph centrality in Lines 30 to 33. However, we believe that these graph centrality measures based on infection subgraph (which only contains infected nodes) are incomplete. Thus, we consider heuristic methods using both infected and uninfected nodes. The experimental results in the manuscript shows that our algorithm could beat centrality methods such as UB in (Comin, C.H., et al. Phys Rev E, 2011) with only infected information. The paper (Ali, S.S., et al., OSNEM, 2020) is a recent progress in centrality-based source identification algorithm and we find it valuable. We add this paper as a reference as well.

Comment 3. While, I don’t doubt the intentions of the authors, however, if I am not wrong, Algorithm 1 in the manuscript, i.e., BLRSI, seems starkly similar to Algorithm 1 (LPSI) of manuscript reference number [25] (Wang, Zheng, et al., AAAI, 2017). The only difference which I see is that LPSI estimates multiple sources by checking whether the label propagation score of a node with respect to its neighbors is the highest and considers it as one of the sources if that is the case. In the algorithm (BLRSI), presented in this manuscript, the highest label score amongst all the observed infected nodes is considered to be the source. I believe this has already been achieved in (Ali, S.S., et. al., A Revisit to the Infection Source Identification Problem under Classical Graph Centrality Measures. OSNEM, 2020), where, for the purpose of comparison, the authors have tuned the same algorithm (LPSI) to capture a single source in, I believe, exactly the same way as the BLRSI algorithm presented in the manuscript. Therefore, my question is what makes BLRSI algorithm different to those algorithms given BLRSI has been proposed/designed (observe the language in the Abstract Section and Line 102) in this manuscript? (Also, the analysis of BLRSI provided in Section “Algorithm analysis of Basic Label Ranking” is quite similar to the analysis of LPSI in Wang, Zheng, et al., AAAI, 2017.

 Response 3. The intentions of our algorithm BLRSI with complete observation are indeed similar to (Wang, Zheng, et al., AAAI, 2017). We tend to measure the proportion of infected nodes surrounding each individual by BLRSI. The analysis of BLRSI is provided for the completeness of our research. Meanwhile, we discuss about the bias elimination in our analysis, which is accustomed to the source identification problem in the loss function (Comin, C.H., et al. Phys Rev E, 2011). Meanwhile, BLRSI is not the only progress of our work. We apply the label ranking in a single source scenario with both complete observation and snapshot observation and present a unified label ranking framework via the underlying semi-supervised learning methods. It makes our algorithms and framework unique among the existing approaches. We have applied several traditional semi-supervised learning loss functions (BLRSI-P1, BLRSI-P2, TSSI) to address the source identification task with complete observation and snapshot. As we mentioned in the manuscript, with the label ranking framework, more semi-supervised learning algorithms, even graph neural networks could be used to obtain the label ranking scores or restore the network status. Thus, our work, including BLRSI and other label ranking algorithms would have profound significance for propagation source identification. 

Comment 4. For complete observation, while the comparison of BLRSI against NetSleuth/SSNS (manuscript ref. no. [12] (Prakash, Aditya B., et al., ICDM, 2012)) is fine, I suggest the comparison should be made against EPA as well (Ali, S.S., et al., CIKM, 2019), given EPA also exploits uninfected nodes to estimate the source of infection. Besides, EPA is a recent work which has been shown to beat NetSleuth/SSNS (Prakash, Aditya B., et al., ICDM, 2012) in single source infection identification scenario.

 Response 4. Since EPA is a recent study which also makes use of uninfected nodes like we do in BLRSI and achieves quite good results, we compare our label ranking algorithms with EPA as well on the 6 datasets. Experimental results can be seen in the revised manuscript. It seems that EPA would be a powerful competitor of our proposed BLRSI. However, better performances of AED and DR on most of the networks make us believe that BLRSI is still an excellent algorithm for source identification. We discuss this in the revised manuscript.

Comment 5. For snapshot observation, all the three proposed methods, VLRSI, TSSI-GFHF and TSSI-LGC should and can (if I am not wrong) at least be compared with one of the recent existing works. While the authors, from Lines 368-373, do argue that the existing works have different approaches in comparison to their own methods and thus the comparison is not suitable, however, in the reasons specified by them, the authors do not consider the methods (for comparison) which plainly consider the snapshot network to estimate the source from. The direct and simple way is to compare the proposed methods (especially VLRSI) with those methods which work on SIR model, for example with Reverse Infection (RI) method (Zhu et. al., Information source identification in SIR model: A sample-path-based approach, IEEE/ACM Trans. on Networking, 2016). In case, this still is not possible, the authors should explain the same.

 Response 5. Noted that we assume the observation vector as O = {OL, OU}, we could observe the exact status (no matter S/I/R or S/I) of a proportion of nodes (OL). This means, we would know the true status of some uninfected nodes as well. For those algorithms designed for SIR model, only infected (I) nodes are observed. We could not distinguish S or R in this scenario, which means no exact information would be given about any uninfected node. Also, the network restoration stage needs at least one known infected and uninfected node as well. It would be unfair to compare with the algorithms for SIR model such as Reverse Infection (RI). Thus, we could only compare VLRSI with Reverse Greedy (Luo W, Tay WP, Leng M. How to identify an infection source with limited observations. IEEE J Sel Top Signal Process, 2014) by setting all observed nodes as infected ones under SIR model (which is not completely accord with our motivation of using both infected and uninfected nodes). Still, we compare these two algorithms in the revised manuscript. But the results could not completely demonstrate the effectiveness of our proposed approach. Meanwhile, we point out that we design this algorithm under the situation that we could sample a proportion of individuals to obtain their exact status, which would be meaningful in real scenario.

 Comment 6. I observe the average degree of the datasets used for experimentation is quite high and, hence, the diameter of these graphs would be on the smaller side and density on the higher side. Therefore, the AED might tend to appear on the good side of things. Lower average degree would mean, higher AED. This is also confirmed by the results (Table 2) achieved on Roget dataset with comparatively lower average degree tending to make AED higher. Typically, if the density of the graph is high, AED becomes small (which makes sense, since the diameter is small in the first place) and DR is not so good. This also explains why DR is so low even on ER graph (26%), when on ER graph source identification performance is generally better. I would suggest authors add density values and diameter values of each dataset in the dataset table (Table 1) and explain this relation (between graph diameter/density and AED/DR).

 Response 6. The suggestion from the reviewer could be quite constructive. The potential relationship between the evaluation metrics of our source identification algorithms and network structures could be revealed from our experimental results. We have added related discussion in part Results and Discussions.

 Comment 7. Generally, in how many iterations does BLRSI achieve convergence in the experimentations that authors have conducted? If there is some iteration number, the authors should add that in the manuscript to help future researchers work with an iterative process instead of a convergent one, wherever suitable.

 Response 7. BLRSI generally takes 4 or 5 iterations to achieve convergence in our experimentations. We have added this result in part Results and Discussions of the revised manuscript.

Comment 8. It appears that the methods proposed in this work can estimate an infection source without having any information on the underlying model of infection propagation. Given the importance of this, I suggest the authors mention this clearly in the manuscript.

Response 8. Our proposed algorithms including BLRSI, VLRSI and TSSI could locate the infection source without knowing the underlying model of infection propagation as the reviewer mentioned. We have pointed out this information in the revised manuscript.

In conclusion, we have carefully considered the comments you made and make revision to our origin work. We appreciate your valuable suggestions and hope you would consider the manuscript acceptable.

Thanks,

Sincerely yours,

Biqing Huang

Professor

Dept. of Automation

Tsinghua University

Beijing 100084, China

Tel: 0086 10 6278 1655

Email: hbq@tsinghua.edu.cn

---

## [Decision Letter · Decision Letter 1]

30 Oct 2020

PONE-D-20-07163R1

Source Identification of Infectious Diseases in Networks via Label Ranking

PLOS ONE

Dear Dr. Huang,

Thank you for submitting your manuscript to PLOS ONE. After careful consideration, we feel that it has merit but does not fully meet PLOS ONE’s publication criteria as it currently stands. Therefore, we invite you to submit a revised version of the manuscript that addresses the points raised during the review process.

We look forward to receiving your revised manuscript.

Kind regards,

M. Sohel Rahman, Ph.D.

Academic Editor

PLOS ONE

Additional Editor Comments (if provided):

Both reports are largely positive, albeit with minor identified issues. Please attend to the comments and resubmit as soon as possible. 

Reviewers' comments:

Reviewer's Responses to Questions

**Comments to the Author**

1. If the authors have adequately addressed your comments raised in a previous round of review and you feel that this manuscript is now acceptable for publication, you may indicate that here to bypass the “Comments to the Author” section, enter your conflict of interest statement in the “Confidential to Editor” section, and submit your "Accept" recommendation.

Reviewer #1: All comments have been addressed

Reviewer #2: (No Response)

2. Is the manuscript technically sound, and do the data support the conclusions?

Reviewer #1: Yes

Reviewer #2: Yes

3. Has the statistical analysis been performed appropriately and rigorously? 

Reviewer #1: Yes

Reviewer #2: Yes

4. Have the authors made all data underlying the findings in their manuscript fully available?

Reviewer #1: Yes

Reviewer #2: Yes

5. Is the manuscript presented in an intelligible fashion and written in standard English?

Reviewer #1: Yes

Reviewer #2: Yes

6. Review Comments to the Author

Reviewer #1: I believe all the comments have been addressed and this manuscript could be considered for acceptance. However, I need the authors to fully address the following points before the paper is published. The respected editor can personally ensure the authors incorporate these few, small changes in their accepted manuscript:

1. As the authors have admitted that BLRSI is indeed similar to LPSI (Wang, Zheng, et al., AAAI, 2017), they must explicitly state this in the manuscript especially when they start discussing BLRSI.

2. The authors have done a commendable job comparing their proposed methods with EPA (Ali, S. S., et.al., EPA: Exoneration and Prominence based Age for Infection Source Identification. CIKM, Nov. 2019). However, the language in lines 421 and 422 in the revised manuscript is not appropriate. While it is noted that EPA’s performance is not as good as the proposed methods on Roget network, the authors should understand that EPA has a good/better performance on French network (on DR), besides being better in Football and BA. Therefore, the authors should remove the generalization in these lines and talk about specific details clearly while noting that EPA outperforms the proposed methods in 3 out of 6 networks (either in DR or AED or both) and provides a stiff competition to the proposed methods in ER graph (on DR), as shown in Table 2.

3. The authors are advised to recheck the time complexity of UB, RC and SNSS (Table 3). As betweenness centrality of all the nodes in a graph involves calculating all the shortest paths between all the pairs of nodes, the complexity should be O(n^3) (see Floyd-Warshall algorithm). Similarly SNSS requires calculating eigenvalues from a matrix which generally has the time complexity of O(n^3). I believe RC on a general graph would have the complexity of O(n^3) as well. Authors need to check this again.

4. In ref. [26] in the revised manuscript, the name of one of the authors is missing. Authors are advised to correct it. Here is the full reference: Ali SS, Anwar T, Rastogi A, Rizvi SA. EPA: Exoneration and Prominence based Age for Infection Source Identification. InProceedings of the 28th ACM International Conference on Information and Knowledge Management 2019 Nov 3 (pp. 891-900).

5. In Supporting Information, "S1 Fig" is missing.

Reviewer #2: The paper “Source Identification of Infectious Diseases in Networks via Label Ranking” examines the coherence and efficiency of label ranking for exploring the possibility of a vertex or node to be the source of infection in a network (represented by graphs). The paper gives detailed comparisons with existing works to prove the superiority of their approach and at the same time, provides a fairly decent result in terms of the time complexity. The manuscript is technically sound, and the data supports the conclusions. Statistical analysis has been performed rigorously and the data has been made fully available. The manuscript is written in standard English and is presented in an intelligible fashion.

It is highly admirable that the paper’s revised version has introduced a number of new experiments as per the suggestions in the primary review. Moreover, what’s commendable is that the study acknowledges that the algorithm design and calculations are simplified due to the network’s abstractions which do not completely visualize all practicalities. The findings in Table 2 are rather impressive given the fact that BLRSI shows better results and satisfactory timings. Although SSNS shows more superiority in terms of timing, the authors have addressed this and shown that BLRSI outperforms SSNS for accuracy. The paper is enriched with lots of relevant comparisons which have made it easier to accredit. Furthermore, the authors have added extensive literature review that allows readers to comprehend the findings in a more convincing manner.

However, some minor issues can be addressed in the first revision. Firstly, the authors have provided Table 5 that analyzes the impact of the initial label vector y. It is seen that omitting the impact of either infected nodes or uninfected ones yields poor results. Although this particular observation is quite intuitive, the other observation where giving higher weights to uninfected nodes or retaining the original assignment like Eq.(1) produces superior results than giving higher weights to infected nodes could use some discussion. That is, why does punishing uninfected nodes give better results than crediting infected ones? This is an unexplored outcome of the experiment and can lead to interesting conversations or further analysis in future works. Basically, the authors could either provide some clarity on this end and explain the reasoning behind it, or perhaps provide a few more entries to the table through a small number of experiments showing that this is not an absolute outcome and will differ for datasets.

Secondly, there is a small issue with Algorithm 2. This algorithm is almost similar to Algorithm 1 with minimal differences. In both these algorithms, k is set to be the parameter that represents the number of iterations. However, although this parameter is incremented in Algorithm 1 (Line 11), Algorithm 2 does not show it. Although it is intuitive that k will be incremented for reaching convergence, having this operation in one algorithm and not having it in another shows inconsistency and can be avoided. The same applies to Algorithm 4.

Thirdly, since Algorithm 2 is almost identical to Algorithm 1 with minor differences (one of them being the definition of the initial label vector) it seems unnecessary to repeatedly show the same operations all over again. Instead, the common algorithm (particularly the iterations) could be set as a subroutine and Algorithms BLRSI and VLRSI could separately call the common subroutine with different parameters. This would also give more insight regarding where exactly the difference between Algorithm 1 and 2 lies. Although this does not seem absolutely necessary, it is something the authors can consider.

In conclusion, the paper provides some interesting observations and acknowledges the limitations. The impressive results in terms of accuracy and time complexity allow readers to comprehend the superiority of this approach.

7. PLOS authors have the option to publish the peer review history of their article (what does this mean?). If published, this will include your full peer review and any attached files.

Reviewer #1: No

Reviewer #2: No

---

## [Author Response · Author response to Decision Letter 1]

2 Dec 2020

Reviewer #1:

Comment 1. As the authors have admitted that BLRSI is indeed similar to LPSI (Wang, Zheng, et al., AAAI, 2017), they must explicitly state this in the manuscript especially when they start discussing BLRSI.

Response 1. We state that we were inspired by source prominence and borrow some ideas from LPSI (Wang, Zheng, et al., AAAI, 2017) while we start discussing our proposed label ranking algorithms.

Comment 2. The authors have done a commendable job comparing their proposed methods with EPA (Ali, S. S., et.al., EPA: Exoneration and Prominence based Age for Infection Source Identification. CIKM, Nov. 2019). However, the language in lines 421 and 422 in the revised manuscript is not appropriate. While it is noted that EPA’s performance is not as good as the proposed methods on Roget network, the authors should understand that EPA has a good/better performance on French network (on DR), besides being better in Football and BA. Therefore, the authors should remove the generalization in these lines and talk about specific details clearly while noting that EPA outperforms the proposed methods in 3 out of 6 networks (either in DR or AED or both) and provides a stiff competition to the proposed methods in ER graph (on DR), as shown in Table 2.

Response 2. We have removed the generalized discussion for the experimental results in Table 2 and talked about the specific performances of these algorithms. We admit that EPA may have a better performance on some of the small networks. But the stability of BLRSI on different networks, especially large-scale networks would make it a better and more practical souce identification algorithm.

Comment 3. The authors are advised to recheck the time complexity of UB, RC and SNSS (Table 3). As betweenness centrality of all the nodes in a graph involves calculating all the shortest paths between all the pairs of nodes, the complexity should be O(n^3) (see Floyd-Warshall algorithm). Similarly SNSS requires calculating eigenvalues from a matrix which generally has the time complexity of O(n^3). I believe RC on a general graph would have the complexity of O(n^3) as well. Authors need to check this again.

Response 3. According to the origin paper of SSNS (Prakash BA, et al. Spotting culprits in epidemics: How many and which ones?), the time complexity of SSNS is indeed O(m). The time cost of our experiments in Table 2 shows that SSNS is quicker than BLRSI, which has the time complexity of O(k*m). Same for RC and UB, they both have the time complexity of O(n^2). Time complexity of more source identification algorithms could be seen in Ref [10] of our manuscript.

Comment 4. In ref. [26] in the revised manuscript, the name of one of the authors is missing. Authors are advised to correct it. Here is the full reference: Ali SS, Anwar T, Rastogi A, Rizvi SA. EPA: Exoneration and Prominence based Age for Infection Source Identification. InProceedings of the 28th ACM International Conference on Information and Knowledge Management 2019 Nov 3 (pp. 891-900).

Response 4. We are truly sorry for missing one of the authors. We have corrected it in the revised manuscript.

Comment 5. In Supporting Information, "S1 Fig" is missing.

 Response 5. We add the S1 Fig into our revised version.

Reviewer #2:

 Comment 1. Firstly, the authors have provided Table 5 that analyzes the impact of the initial label vector y. It is seen that omitting the impact of either infected nodes or uninfected ones yields poor results. Although this particular observation is quite intuitive, the other observation where giving higher weights to uninfected nodes or retaining the original assignment like Eq.(1) produces superior results than giving higher weights to infected nodes could use some discussion. That is, why does punishing uninfected nodes give better results than crediting infected ones? This is an unexplored outcome of the experiment and can lead to interesting conversations or further analysis in future works. Basically, the authors could either provide some clarity on this end and explain the reasoning behind it, or perhaps provide a few more entries to the table through a small number of experiments showing that this is not an absolute outcome and will differ for datasets.

 Response 1. The detailed impact of the initial assignment of label vector is an extensive problem of our study. We think that different properties of contact network and infection subgraph might lead to a different optimal label assignment. Since giving the same weight to infected and uninfected nodes could perform well, even if it’s not the best choice sometimes, we believe researchers could try this way as specific characteristics of the target source identification problem is always unclear. More concrete experiments could be conducted to assess this issue in the future.

 Comment 2. Secondly, there is a small issue with Algorithm 2. This algorithm is almost similar to Algorithm 1 with minimal differences. In both these algorithms, k is set to be the parameter that represents the number of iterations. However, although this parameter is incremented in Algorithm 1 (Line 11), Algorithm 2 does not show it. Although it is intuitive that k will be incremented for reaching convergence, having this operation in one algorithm and not having it in another shows inconsistency and can be avoided. The same applies to Algorithm 4.

Comment 3. Thirdly, since Algorithm 2 is almost identical to Algorithm 1 with minor differences (one of them being the definition of the initial label vector) it seems unnecessary to repeatedly show the same operations all over again. Instead, the common algorithm (particularly the iterations) could be set as a subroutine and Algorithms BLRSI and VLRSI could separately call the common subroutine with different parameters. This would also give more insight regarding where exactly the difference between Algorithm 1 and 2 lies. Although this does not seem absolutely necessary, it is something the authors can consider.

Response 2&3. We find these two comments valuable and reorganize Algorithm 1 and Algorithm 2 in our revised version. This might improve the readability of our paper.

---

## [Decision Letter · Decision Letter 2]

29 Dec 2020

Source Identification of Infectious Diseases in Networks via Label Ranking

PONE-D-20-07163R2

Dear Dr. Huang,

We’re pleased to inform you that your manuscript has been judged scientifically suitable for publication and will be formally accepted for publication once it meets all outstanding technical requirements.

Kind regards,

M. Sohel Rahman, Ph.D.

Academic Editor

PLOS ONE

Additional Editor Comments (optional):

Reviewers' comments:

Reviewer's Responses to Questions

**Comments to the Author**

1. If the authors have adequately addressed your comments raised in a previous round of review and you feel that this manuscript is now acceptable for publication, you may indicate that here to bypass the “Comments to the Author” section, enter your conflict of interest statement in the “Confidential to Editor” section, and submit your "Accept" recommendation.

Reviewer #2: All comments have been addressed

2. Is the manuscript technically sound, and do the data support the conclusions?

Reviewer #2: Yes

3. Has the statistical analysis been performed appropriately and rigorously? 

Reviewer #2: Yes

4. Have the authors made all data underlying the findings in their manuscript fully available?

Reviewer #2: Yes

5. Is the manuscript presented in an intelligible fashion and written in standard English?

Reviewer #2: Yes

6. Review Comments to the Author

Reviewer #2: (No Response)

7. PLOS authors have the option to publish the peer review history of their article (what does this mean?). If published, this will include your full peer review and any attached files.

Reviewer #2: No

---

## [Editor Report · Acceptance letter]

4 Jan 2021

PONE-D-20-07163R2 

Source Identification of Infectious Diseases in Networks via Label Ranking 

Dear Dr. Huang:

I'm pleased to inform you that your manuscript has been deemed suitable for publication in PLOS ONE. Congratulations! Your manuscript is now with our production department. 

Kind regards, 

on behalf of

Dr. M. Sohel Rahman 

Academic Editor

PLOS ONE